# A Quantitative Social Network Analysis of the Character Relationships in the Mahabharata

**Eren Gultepe * and Vivek Mathangi**

Department of Computer Science, Southern Illinois University Edwardsville, Edwardsville, IL 62026, USA
* Correspondence: egultep@siue.edu

**Abstract:** Despite the advances in computational literary analysis of Western literature, in-depth analysis of the South Asian literature has been lacking. Thus, social network analysis of the main characters in the Indian epic *Mahabharata* was performed, in which it was prepossessed into verses, followed by a term frequency–inverse document frequency (TF-IDF) transformation. Then, Latent Semantic Analysis (LSA) word vectors were obtained by applying compact Singular Value Decomposition (SVD) on the term–document matrix. As a novel innovation to this study, these word vectors were adaptively converted into a fully connected similarity matrix and transformed, using a novel locally weighted K-Nearest Neighbors (KNN) algorithm, into a social network. The viability of the social networks was assessed by their ability to (i) recover individual character-to-character relationships; (ii) embed the overall network structure (verified with centrality measures and correlations); and (iii) detect communities of the Pandavas (protagonist) and Kauravas (antagonist) using spectral clustering. Thus, the proposed scheme successfully (i) predicted the character-to-character connections of the most important and second most important characters at an F-score of 0.812 and 0.785, respectively, (ii) recovered the overall structure of the ground-truth networks by matching the original centralities (corr. $> 0.5$, $p < 0.05$), and (iii) differentiated the Pandavas from the Kauravas with an F-score of 0.749.

**Keywords:** natural language processing; literary heritage; word vectors; social network analysis; character networks; *Mahabharata*

## 1. Introduction

Nineteenth-century English literature has been of particular interest for the application of NLP techniques. Authors such as Arthur Conan Doyle, Charles Dickens, and Charlotte Brontë have had their works extensively analyzed in traditional literary arts and in digital humanities [1]. For example, the works of Jane Austen (*Pride and Prejudice, Persuasion, Mansfield Park*, etc.) [2] have been thoroughly examined with NLP techniques and have led to novel discoveries [3,4]. These studies demonstrated that a significant amount of words used by Austen in her books are unique, providing evidence for clear differences between men and women in their actions and how they conversed, and for the time period, her works are very emotive.

Grayson et al. [2] used word-embedding techniques to show a macro-level organization of 19th-century novels. They used 12 works from various 19th-century English novelists as the corpus and utilized Word2Vec for generating the word vectors of the characters in each novel. They conclude that word vectors generated with their own variant of Word2Vec called Novel2Vec successfully give an outline of the novels in the corpus. The authors assert that similar performance can be expected on other books from that period. Additionally, social network analysis has been popularly utilized to reveal significant information such as alliances, genealogy, status, etc. from the social dynamics. For example, Agarwal et al. detected and classified social events in *Alice in Wonderland* using social network analysis [5].

In 19th-century English literature, social networks have been created for the main character groups of many literary works such as *Our Mutual Friend* (Charles Dickens), *Alice's Adventures in Wonderland* (Lewis Carroll) and *A Study in Scarlet* (Arthur Conan Doyle), to name a few [1,5,6]. Social network analysis is frequently utilized for inferring character information such as importance and influence [7]. For instance, Elson et al. [1] generated their social networks using almost 50 novels based on the dialogue between characters. In their social networks, the edges between nodes are instances of bilateral conversation between characters and the edge weight depends on the frequency of those conversations. In their analysis, they utilized graph density and balance measures, among others, in inferring information about community cohesion and the interconnectedness of relationships. Finally, Elson et al. [1], quantitatively demonstrate, counter to traditional understandings, that the narrative style, whether first person or third person, is a stronger indicator than rural vs. urban setting of social cohesion in novels.

For Western literature, the above studies only represent a small subset of the previous and ongoing work, whereas in non-Western works, particularly in the South Asian literature, there is a lack of studies. Surprisingly, research on the Indian literary tradition has been very sparse, even for prominent works such as the *Mahabharata* (Figure 1). In *A Computational Analysis of the Mahabharata*, Das et al. [8] provide one of the more visible applications of NLP techniques and analysis to the epic. In contrast to the LSA procedure employed in this study, Das et al. opted for the much simpler and straightforward procedure of co-occurrence analysis to generate their social network. While not as powerful as the LSA, co-occurrence analysis was sufficient for their experiments, in which they analyzed the negative or positive sentiment of the individual characters. Their work corroborates known interpretations of the significant characters in the *Mahabharata*, as well as literary properties such as tone and plot.

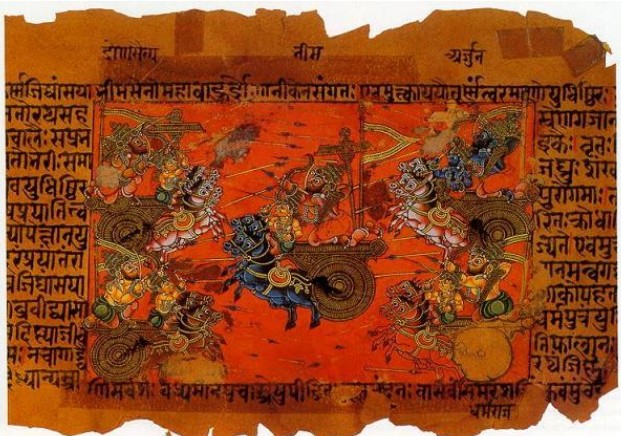

**Figure 1.** Depiction of the Kurukshetra War in the *Mahabharata*. An illustration from a Sanskrit manuscript of the *Mahabharata* showing the Kurukshetra War. This is considered climax of the *Mahabharata*, where the Pandavas and the Kauravas battle for the throne of Hastinapur. Possibly based on an ancient, historical war around 1000 BCE, this battle represents a major section of the *Mahabharata* with multiple chapters (parvas) being dedicated to it. Source: adapted from [9].

*Research Aim*

Computational literary analysis of the Western literature has benefited greatly from numerous amounts of technical innovation [10,11] and novel insights such as informational retrieval, summarization and machine translation [12]. In comparison to its Western counterpart, Eastern and South Asian literature has been critically understudied. This study seeks to initiate the analysis of non-Western literary works and to more justly represent the many rich and diverse cultures of humanity, with the goal of uncovering the hidden knowledge and potential insights that have been subsequently neglected.

This study provides an extensive literary analysis of the *Mahabharata* using machine learning techniques and social network methods. It is focused on understanding the intrapersonal and interpersonal relationships among the heroes, royalties, and gods of Hindu mythology. Understanding these relationships will lead us to the goal of corroborating known facts about these mythologies and understanding how they have evolved over time, and reveal insights and unexpected interactions that would be difficult to observe without data-driven techniques. The significance of this work lies in developing a catalogue of techniques and perspectives to provide a modern and nuanced understanding of our shared cultural heritage, not just for Indian culture, but also for other major world cultures. Specifically, the research presented herein analyzes what is overlooked in the traditional analysis of the *Mahabharata* and what can be objectively supported once that information is quantified. Furthermore, it investigates how data representation and data structure can influence the perspectives of the characters.

This research is an expansion in scope of traditional Western novels, which are usually 300–400 pages [13], whereas the corpus being analyzed is many times larger at 13,000 pages and 2000 chapters. Additionally, this study considers a large roster of 20 main characters being analyzed in a relatively larger and more realistic social network. The three primary investigations are: (i) social network construction using word vectors to create a locally weighted KNN graph, (ii) social network analysis for understanding the character relationships by way of centrality measures and recovery of character-to-character connections, and (iii) community detection with spectral clustering [14] to determine the two primary groups traditionally represented in the *Mahabharata* as the Pandavas (righteous/protagonist) and the Kauravas (evil/antagonist).

## 2. Materials and Methods

The following outlines the three main steps in the pipeline: (i) corpus and text preprocessing leading up to the generation of the LSA word vectors and the construction of the ground truths, (ii) the construction of the social network using LSA word vectors, co-occurrence and manual annotation, and (iii) social network analysis using centralities, character analysis with an emphasis on relationships and community detection.

### 2.1. Processing Pipeline

First, in the preprocessing phase, the full text of the *Mahabharata* and the corresponding summary text, the *Penguin Companion to the Mahabharata*, are divided into books (parva) and each respective book is divided into sections. Thus, each section is a document in the TF-IDF computation, which results in the term–document matrix. The final representation of the characters is in word vector form and is achieved via the LSA, specifically, truncated SVD (see Section 2.2.2). Additionally, the ground truths serve as the control in the social network analysis experiments. The ground truths were compiled from the character indexes in the *Penguin Companion to the Mahabharata* and are represented as a matrix of connections (i.e., adjacency matrix) between the characters.

The social networks in the analyses are obtained in three ways (Figure 2). The word vector network is obtained from a fully connected similarity matrix that is constructed using the cosine similarities of the word vectors and by adapting the $k$ parameter in the Gaussian function (see Algorithm 1) to the frequency of the character names. From this similarity matrix the $k$ shortest distances are preserved, adaptively, resulting in the locally weighted KNN graph that represents the character social network. In the case of the co-occurrence social network, co-occurrence of the characters is obtained from the full text of the *Mahabharata*. Specifically, the incidence of characters within each sentence is compiled into a co-occurrence matrix, which is converted to the social network. Finally, in the case of the annotated social network, the connection matrix from the ground truth is converted to a graph representing the ground truth social network.

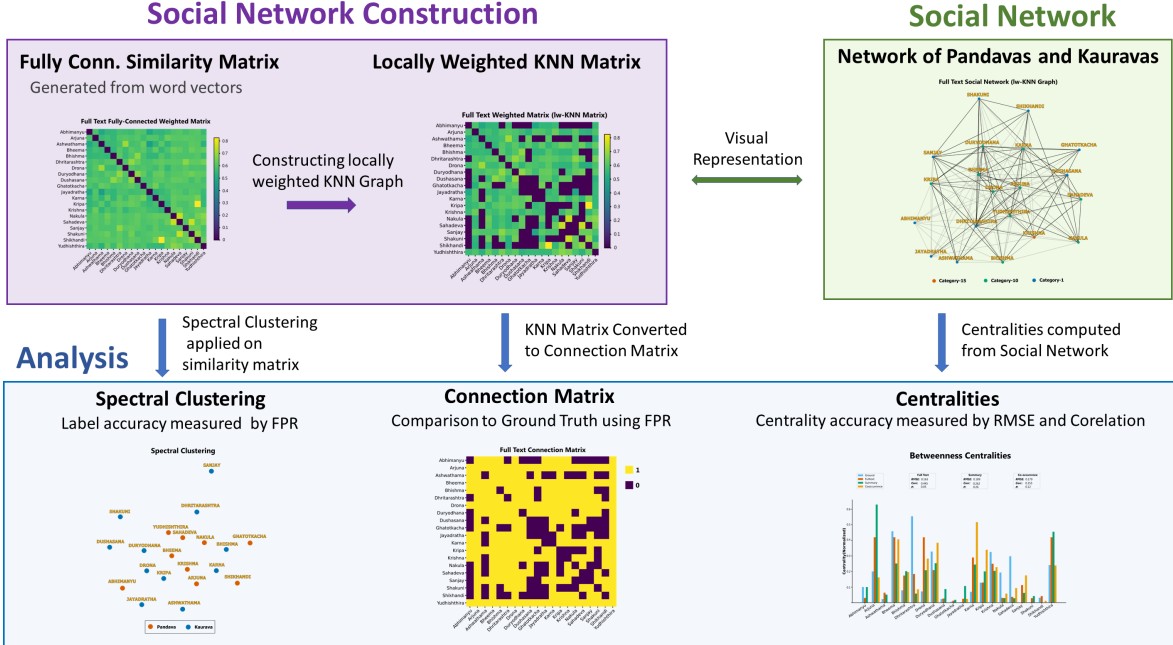

**Figure 2.** Overall pipeline for social network analysiswork Analysis. From the word vectors, the fully connected similarity matrix is computed and is used directly with spectral clustering for community detection of the Pandavas and Kauravas. For analyzing individual character social networks, the locally weighted KNN (lw-KNN) matrix is formed by dropping out all $k + 1$ neighbours and beyond, where $k$ is determined by a name's frequency. Also, the lw-KNN matrix is binarized to form a connection matrix and used in determining the recovery of the character-to-character relationships.

Three main experiments are utilized for the social network analysis. In the first experiment, the character-to-character relationships in the word vector social network and the co-occurrence social network are each compared to the ground truth social network. Here, the F-scores are used to evaluate the accuracy of the relationships captured by the word vector and co-occurrence networks. In the second experiment, in both constructed social networks, the following centrality measures are computed: betweenness, closeness, degree, and eigenvectors. Then, RMSE and Pearson correlation measures are calculated with respect to the centralities obtained from the ground truth social network. In the final experiment, community detection is performed using spectral clustering, which is applied on the fully connected similarity matrix that underlies the word vector social network. Then, the obtained two clusters representing the opposing groups are formed, specifically the protagonists (Pandavas) and antagonists (Kauravas). These two clusters are then compared to the ground truth protagonist and antagonist groups using the F-score measures.

*2.2. Corpus and Text Preprocessing*

This research uses K. M. Ganguli's translation of the *Mahabharata* [15] obtained from *Sacred Texts* [16]. The size and scope of the *Mahabharata* is many times larger than the average Western novel. The corpus consists of 130,000 pages that are split into volumes known as Parvas, which can be further sub-divided into *verses*. This massive amount of textual data is further complimented by a large 300,000 unique vocabulary. Thus, careful pre-processing is carried out to ensure the suitability and reliability of data.

Firstly, word tokenization is performed, removing all special symbols and characters as well as words less than 3 letters in length. Subsequently, common stop-words from the standard NLTK and archaic stop-words, such as 'hath' and 'thou', are filtered out. Detokenization is then performed recomposing all the sections of the corpus. After this, the corpus is prepared directly for the LSA process.

A specific set of main characters, outlined in Table 1, is essential for this research as all subsequently analysis will be derived from the word vector representations associated with

them. This list of characters is compiled from the character summaries of the *Penguin companion to The Mahabharata* [17]. The criteria in choosing the main characters are their importance to the overarching narrative, the relative status, or significance of the character, which is quantified by their word count/frequency and their distribution throughout the book.

**Table 1.** The main Pandavas and Kauravas in the *Mahabharata*.

| Pandavas (Label 1) | Kauravas (Label 0) |
|---|---|
| Abhimanyu | Ashwathama |
| Arjuna | Bhishma |
| Bheema | Dhritarashtra |
| Ghatotkacha | Drona |
| Krishna | Duryodhana |
| Nakula | Dushasana |
| Sahadeva | Jayadratha |
| Shikhandi (Amba) | Kripa |
| Yudhishthira | Sanjay |
| | Shakuni |

The final preparation is computing the word count of each individual character, which in addition to approximating the above criteria of each the character, also reflects the quantity of relations the character should have. Here, characters with more significance and thus a higher word frequency of mentions of their names will have considerably more relations in contrast to other characters. This idea will be applied to the concept of bands in the locally weighted KNN graph, when analyzing the similarity of characters concurrently.

### 2.2.1. Establishing Ground Truths

In preparing a suitable and valid representation for the ground truths, the *Penguin Companion to the Mahabharata* [17] contains an outline and summary of the plot of the Mahabharata and, important to this research, an informative character index which is heavily referenced. The ground truth connection matrix (also known as the adjacency matrix), which represents the set of all relationships for all the characters, is compiled directly from the character index. Here, the character index provides summaries for all the characters, which explicitly indicates how the characters interact with each other. From this, all the characters' relationships are recorded and tabulated as the connection matrix. In this format, 1 indicates an existing relationship and 0 indicates no relationship (Figure 3A).

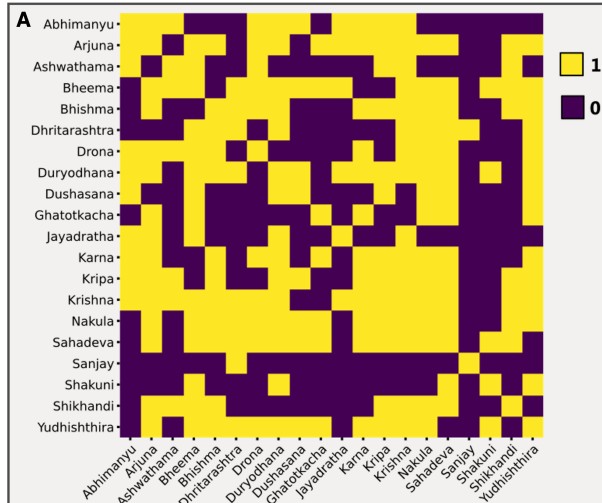
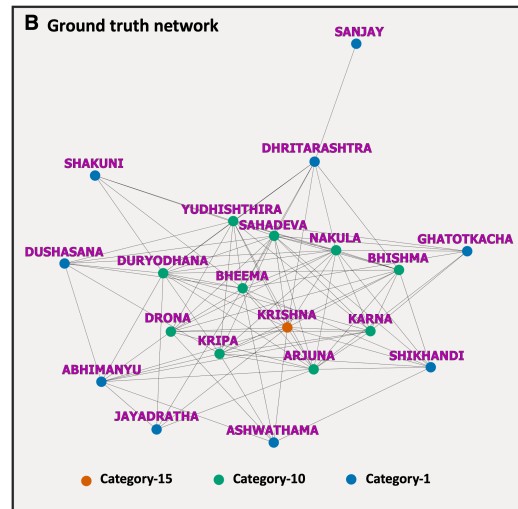

**Figure 3.** Ground truths for social network analysis. (**A**) Connection matrix showing the true character-to-character relationships for each character. (**B**) Social network representation of the character-to-character relationships.

Similarly, community detection is applied for the discovery of characters who belong to either the Pandavas or Kauravas groups. The character summaries in the character index specify the faction of the character as a Pandava or Kaurava. Thus, the character labels are generated for the list of main characters, with a 1 indicating a Pandava and a 0 indicating a Kaurava. This set up is summarized as Table 1.

### 2.2.2. Word Vectors

Word vectors are robust embeddings of word meanings, which can represent a matrix of one-hot encodings or bag-of-words as a compressed matrix. Subsequently, recent advances have utilized neural nets such as Word2Vec and GloVe in generating a standard set of high performing word vectors [18,19]. This investigation, however, uses LSA via SVD, which out-competes the neural net methods for smaller and specialized corpuses [20] and is on par with Word2Vec and GloVe in larger datasets [21].

After the preprocessing phase, the corpus contains over 2000 'cleaned' chapters. Subsequently, the corpus is divided by verses and organized as a document–term matrix, where each row represents a verse and the columns represent the unigram words in the vocabulary of the *Mahabharata*. The TF-IDF is calculated for each entry of this document–term matrix with the following stipulations: no constraints for the amount of vocabulary, but with the condition that the maximum IDF is 0.8. Additionally, the smooth IDF is calculated whereby 1 is added to every frequency to avoid division by zero.

Following the TF-IDF transformation, compact SVD is performed to produce the word vectors for each term in the vocabulary, in which the names of the characters are also included (Figure 4). As shown in Equation (1), using only the top $d$ singular values, the document–term matrix $\mathbf{A}_d \in \mathbb{R}^{v \times n}$, where $v$ is the vocabulary terms and $n$ is the number of verses, is decomposed as the word embeddings $\mathbf{U}_d \in \mathbb{R}^{v \times d}$, singular values $\mathbf{\Sigma}_d \in \mathbb{R}^{d \times d}$, and document embeddings $\mathbf{V}_d^T$,

$$\mathbf{A}_d = \mathbf{U}_d \mathbf{\Sigma}_d \mathbf{V}_d^T. \tag{1}$$

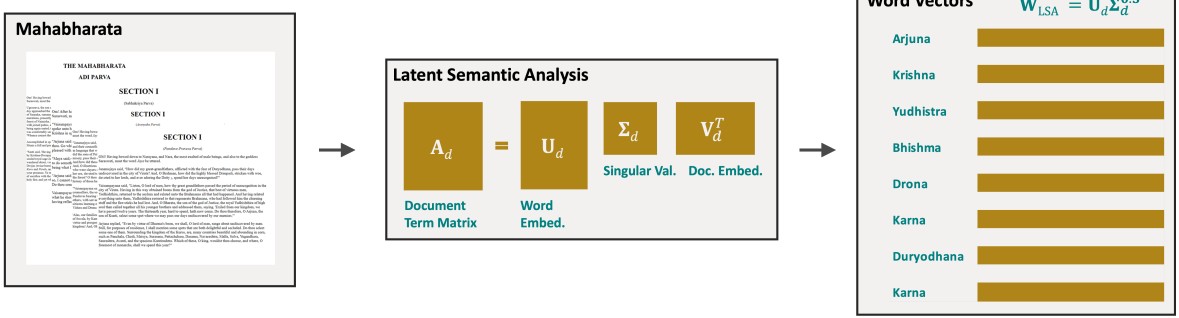

**Figure 4.** Word vector processing pipeline. The pipeline shows all intermediary steps for extracting word vectors from the raw *Mahabharata* text. First, the *Mahabharata* is naturally organized into parvas (verses) and adi-parvas (sub verses) which are partitioned to obtain separate documents. These documents are organized into a document matrix **A** which undergoes TF-IDF processing, generating a document–term matrix. Second, LSA is performed on the document–term matrix using compact SVD and LSA word vectors of the characters names are obtained.

As given in Equation (2), the LSA word vectors are defined as the rows of,

$$\mathbf{W}_{\text{LSA}} = \mathbf{U}_d \mathbf{\Sigma}_d^{0.5}, \tag{2}$$

where scaling the singular values by a power of 0.5 and row normalization to unit length are known to improve representative power [21].

*2.3. Social Network Construction*

A social network is a network structure with entities (characters) as the nodes and edges, which captures the relationships between entities within a social setting [22,23]. Social networks have been generated in a variety of ways using textual information obtained from word occurrence, quoted speech, and eventually more abstract structures such as social events [7]. Analysis of a network's structural properties can be carried out at different relationship levels with the aid of metrics such as centrality measures [24] to reveal relational information such as the influence between characters [25]. As such, social networks can be used to find and justify known and novel information of characters from the social context.

For all three types of social networks (locally weighted KNN, co-occurrence, and ground truth graphs) of the characters in the *Mahabharata*, the networks are organized to minimize the stress function (also known as the energy function) with respect to multidimensional scaling [26]. Essentially, the most similar nodes are organized centrally to maximize their similarity to all other nodes, and the most dissimilar nodes tend to be pushed the edges to indicate their lack of similarity to the other nodes.

2.3.1. Locally Weighted KNN Graph Social Network

K-Nearest Neighbors (KNN) graphs are traditionally used for clustering problems [27,28] and they have been especially productive in fields of data mining [27] and machine learning [29]. Formally, a KNN graph is a specific graph, $G = (V, E)$, where $V$ is the set of node and $E$ the set of edges, where the distance $d = |e\ \epsilon\ E|$ between a node $n\ \epsilon\ V$ to another is within the $k$-th smallest distances from node $n$ to any other in the set $V$ [30]. These experiments thus utilize the properties of these special graphs for determining quantitative social metrics, discussed in the Experimental Setup section, by constructing a weighted KNN graph.

The weighted KNN graph finds the closest associates for each character. This is based on the principle that characters in similar contexts, here quantified by cosine distance, will aggregate together. However, the traditional KNN method has a critical limitation in these experiments; a traditional algorithm will either force an unjustifiable amount of relationships on certain characters or underrepresent the amount for others. To circumvent this limitation, the standard KNN procedure is modified to accommodate the differences in character relations.

The following describes how the standard KNN procedure is modified to mitigate the issue of the mismatch of $k$ to the actual connections a character may have, thereby forming the locally weighted KNN (lw-KNN) graph. First, a fully connected similarity matrix is built from first principles starting from a square-form pairwise distance of characters. Subsequently, a word frequency check is performed on characters to determine the appropriate $k$ value. The characters with word counts higher or lower than a certain range are assigned to the appropriate band, providing a more realistic approximation of the number of relations. The three bands will correspond to three $k$ parameters, particularly: 15, 10 and 5, respectively. Equation (3) shows the word count associated with the full text and Equation (4) shows the word count associated with the summary. Then, the radial basis function is applied to each entry with the appropriately scaled $\sigma$ parameter, which is also dependent on $k$. Then, for each row (character) the $k+1$ neighbor and beyond are filtered out, leaving only $k$ neighbors, and thus providing the locally weighted KNN matrix/graph.

$$k = \begin{cases} 15, & 1500 < count. \\ 10, & 500 < count < 1500. \\ 5, & count < 500. \end{cases} \tag{3}$$

$$k = \begin{cases} 15, & 50 < count. \\ 10, & 10 < count < 50. \\ 5, & count < 10. \end{cases} \tag{4}$$

The $\sigma$ parameter is computed by taking the mean of the *k*-th largest distances for each character. The $\sigma$ value greatly affects the performance of both the fully connected similarity matrix that is used in spectral clustering algorithms [14] and the locally weighted KNN graph used in constructing and determining the character social networks. Algorithm 1 below outlines these steps:

---

**Algorithm 1:** lw-KNN procedure

---

**Input:** Word Vectors as Matrix.
**Output:** Modified KNN Matrix.
1.     Compute Pairwise Cosine Similarity.
2.     Square-form and Sort Distances.
3.     **for** *All Rows* **do**
4.         Obtain current *k* value (based on word count).
5.         Keep first *k* entries.
6.         Compute $\sigma \leftarrow$ Mean of First. *k* Values
7.         Set *k*+1 entries to 0.
8.         Element-wise division by $\sigma$.
9.     Apply Radial Basis Function (Gaussian).

---

The above algorithm provides the basic brute force KNN approach we use for creating a KNN graph with locally weighted edges.

### 2.3.2. Co-Occurrence-Derived Social Network

For the co-occurrence networks, the frequency of typically two words that occur together in a text corpus is used to form the network [31]. Co-occurrence here may refer to the words occurring within the same sentence, paragraph, article, and possibly other frames of co-occurrence windows [32,33]. Co-occurrence methods such as co-occurrence networks are used in diverse fields from biology and ecology [34,35] to more traditional NLP research [36,37]. Here, co-occurrence networks provide a visual representation of relationships derived from the co-occurrence analysis, after which social network analysis can be utilized (see the Experimental Setup section) to derive information.

In this investigation, co-occurrence analysis is applied to the *Mahabharata* text corpus. Using the roster of characters outlined in Table 1, the co-occurrence of characters within each sentence is computed and then used to construct a social network, where each edge is weighted by the number of co-occurrences. This approach is very similar to that used by Das et al. [8] in their construction of the *Mahabharata* social network, allowing for a comparison of the two networks. In our social network analysis, the co-occurrence network is the baseline method against which our lw-KNN graph is compared.

### 2.3.3. Ground Truth Social Network

The ground truth network reflects the social structure of a given literature. This is usually facilitated by character annotations derived from the corpus [1,5]. The section Establishing Ground Truths has already outlined the process of the generation of the ground truth connection matrix, which represents these characters and their social relations. Thus, the ground truth social network for the *Mahabharata* is easily obtained by converting the connection matrix to a graph (Figure 3B).

### 2.4. Social Network Analysis

Social networks are an example of a structure that is fundamental to the point of being ubiquitous in many fields [23]. For the agents (nodes) in the network, social network analysis can reinforce and reveal social dynamics such as relationships and hierarchy [7]. The construction of the social network in the previous sections is a critical step in character analysis as it contains contextual information in a compact form and is readily comparable to that of the ground truth. Additionally, the lw-KNN matrix and spectral clustering

technique provide a means for comparing against the ground truth in order to verify the constructed social networks.

### 2.4.1. Individual Character-to-Character Relationships

The lw-KNN matrix generated from word vectors represents the set of relationships each character has with the other characters. Subsequently, each element within a row quantifies the relationship one character has with another. Then, all non-zero values in a row indicate a measurably significant relation. Thus, it can be assumed that the social network obtained by the lw-KNN matrix is underlain by a connection matrix. Here, the locally weighted KNN matrix is essentially equivalent to a weighted connection matrix. Subsequently, with respect to each technique, the character-to-character relationships are embedded in the connection matrix (ground truth), the KNN matrix (word vectors), and the co-occurrence matrix (co-occurrence analysis).

The validity of the predicted connections by the lw-KNN matrix is determined by comparing each character's predicted connections to other characters against the ground truth connections provided by the social network (see the section Establishing Ground Truths for the compilation of the ground truth relationships). Here, the ground truth represents the actual relationships of each character in a connection matrix. Subsequently, comparing a row of the connection matrix to the corresponding row of the ground truth for all the rows provides a method to measure the accuracy.

To evaluate if the correct connections have been obtained, for each character, their recall (R, i.e., how many of the connections are predicted/selected), precision (P, i.e., of the selected connections, how accurate are the predicted/selected connections), and F-score (F, the harmonic mean of the precision and recall) metrics are computed. For assessing the overall performance of a specific character, the F-score is the metric used since it combines both of the other two metrics. The reported metrics are averaged across all the characters (also known as macro-averaging) for each processing stream.

This is carried out for three different character groupings. A group containing all characters with at least one relationship, i.e., all individuals, is called category 1. Likewise, category 10 and category 15 are for those characters with at least 10 and 15 relationships, respectively. An additional experiment in which the macro-averaged metrics are computed for the characters organized into the Pandavas or Kauravas groups was conducted.

### 2.4.2. Social Network Structure with Centralities

For all the social networks (the lw-KNN graph, co-occurrence, and ground truth), a descriptive set of standard metrics in graph theory, known as centralities, was calculated for each node in the network:

- Betweenness centrality—the extent to which a node is in the shortest path between all other nodes [38].
- Closeness centrality—a calculation of the inverse distance of the shortest part to other nodes [39].
- Degree centrality—the same as the number of edges a node possesses [40].
- Eigenvector centrality (normalized)—the number of nodes with high centrality scores connecting to the target node [41].

For each of the four centrality measures, the root mean squared error (RMSE) and the correlation were calculated for each of the two constructed networks and the ground truth network. Centralities measures are able to convey information about a network's structural properties to reveal relational information such as the influence between characters [24,25].

### 2.4.3. Community Detection with Spectral Clustering

Community detection has been used for identifying groups in large-scale graphs [42–44]. This has been facilitated by spectral clustering methods, which have been shown to have high performance [44]. Spectral clustering has been proven to be very effective and accurate at clustering tasks [14,45]. Spectral clustering is able to deal with

high dimensional input and is not inhibited by local optima [45], making it suitable for a wide range of tasks [46]. The performance of spectral clustering is greatly dependent on the similarity graph and the parameters associated with the similarity measure [14]. In these experiments, the spectral clustering is performed with a similarity matrix that is generated in an adaptive manner. One of the main parameters in its generation, the $\sigma$ parameter, as seen in Algorithm 1, greatly effects clustering [14] and is chosen for each character based on its word frequency as described in the locally weighted KNN graph section. The Shi-Malki algorithm is used here for its efficiency, performance, and reliability [47,48]. Algorithm 2 for this spectral clustering is given below:

---

**Algorithm 2:** Shi-Malik spectral clustering method

---

**Input:** Similarity Matrix. **Output:** Clusters.
1. Compute similarity graph and adjacency matrix **W**.
2. Compute unnormalized Laplacian **L** on **D** − **W**.
3. Compute first $k$ eigenvectors $\mathbf{u}_1, \ldots, \mathbf{u}_k$ of the generalized eigenproblem $\mathbf{Lu} = \lambda \mathbf{Du}$.
4. Let **U** be the column matrix containing the eigenvectors.
5. Cluster rows of **U** with the discretization algorithm [48].
6. Return clusters.

---

The above algorithm essentially performs matrix factorization using generalized eigen decomposition on the graph structure contained in **W**. Then, the reduced set of eigenvectors is used for finding the grouping of observations. For our current study, community detection using spectral clustering has been implemented for identifying which of the two factions (Pandavas or Kauravas) each character in the *Mahabharata* belongs to. The group membership of the characters provided by spectral clustering is compared to the ground truth labels described in the Establishing Ground Truths section.

Similar to the character-to-character relationships, the performance of the predicted faction labels is assessed using the macro-averaged (i.e., averaged across the two factions) F-score, precision, and recall. However, in this context, the recall defines how many of the characters are predicted/selected for each faction, and precision defines, for the selected characters, how accurate the predicted/selected characters are. For clustering algorithms, predicted labels are agnostic to the predicted partitioning of the data. To this end, the Hungarian algorithm is applied in labelling the Pandavas and Kauravas. The Hungarian algorithm provides an efficient solution to what is known as the "assignment problem", i.e., assigning labels to tasks such that an optimal payoff is obtained [49]. In this case, the labels of Pandavas and Kauravas are assigned such that the maximum F-scores are obtained.

### 2.5. Experimental Setup

The experiments are designed to assess the ability of the lw-KNN social networks to capture character-to-character information from the full and summary texts. An overview of the analysis procedure is as follows:

1. LSA word vectors are obtained by varying the dimension from 100 to 1000 for the full-text corpus and 10 to 100 for the summary text corpus.
2. From each set of word vectors in both texts, a social network is constructed.
3. Similarly, a social network using co-occurrence analysis of the full text is constructed.
4. For all constructed social networks, the following are computed:
   - For each character in a network, their character-to-character F-score, precision, and recall are computed with respect to the ground truth social network.
   - For all nodes in a network, the four different types of centralities are computed. Then, for each centrality, the overall RMSE and correlation (with $p$-value) between the constructed network and the ground social network are computed.

To specifically analyze the Pandavas and Kauravas character groups in the *Mahabharata* using the LSA word vectors from full text and summary text for the lw-KNN graphs, the following is performed:

1.    The character-to-character performance metrics are computed for each group against the ground truth social network.
2.    The performance of the detected groups using spectral clustering is computed against the ground truth Pandavas and Kauravas groups.

### 3. Results

*3.1. Social Networks*

Figure 3B shows that in the ground truth social network, the social context encloses the god Krishna (since it has a high number of connections with other characters) within a middle ring of protagonists (Pandavas) and antagonists (Kauravas) and an outer ring of minor/supporting characters (such as Drona). The centrality measures and character-to-character connections of this ground truth social network are used to assess the fidelity of the other constructed social networks.

Although for each LSA word vector a full-text lw-KNN social network is obtained, in Figure 5A, the best performing social networking for character-to-character interactions is displayed from Table 2, which utilizes the vector dimension of 600. The figure shows that the nodes of the full-text social network are arranged markedly based on social significance with characters that are prominent in the story taking more central positions, despite possibly having less character-to-character connections, which is due to the semantic information embedded in the LSA word vector of the character names. This is illustrated with Arjuna being more central than Krishna, despite the latter having more connections, as the former is the main protagonist (the most significant Pandava) of the plot.

For the summary text lw-KNN social network in Figure 5B, the network obtained using an LSA word vector dimension of 50 is displayed, as it was the best performing dimensionality. Here, in terms of a visual inspection, the summary text network appears more similar to the ground truth social network than the full-text network, as it is seen that Krishna is centrally located and the lower connected nodes are more on the periphery.

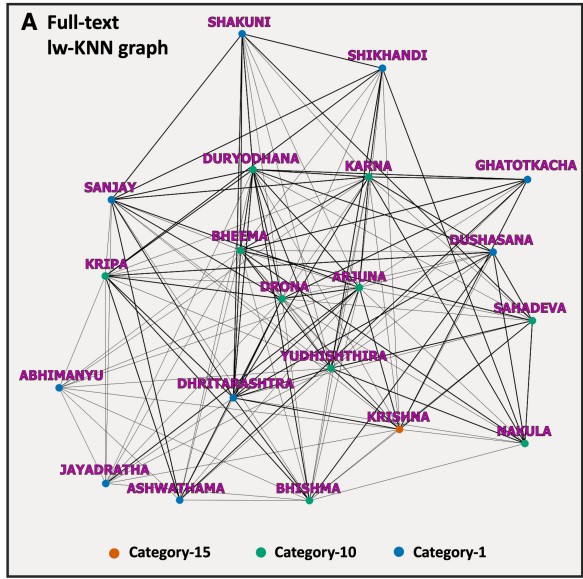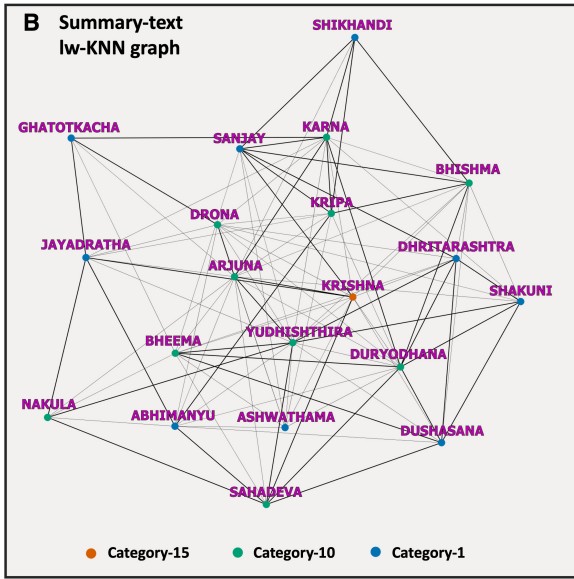

**Figure 5.** *Cont.*

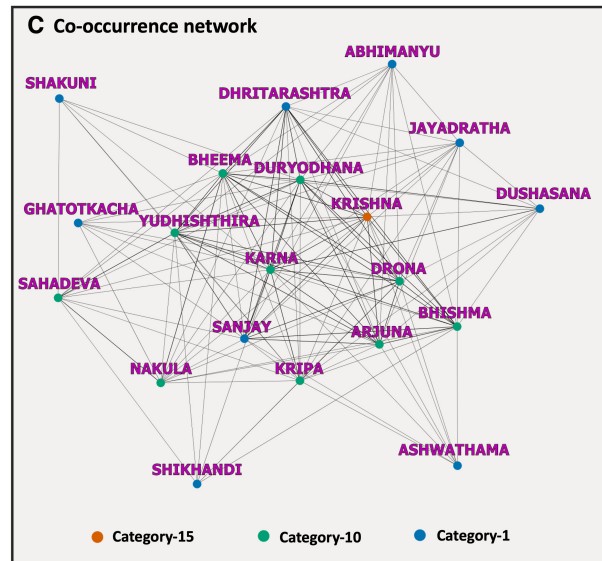

**Figure 5.** Computed social networks of the *Mahabharata*. (**A**) Full-text lw-KNN social network obtained using LSA word vector dimension of 600. (**B**) Summary-text lw-KNN social network obtained using LSA word vector dimension of 50. (**C**) Co-occurrence-based social network developed on the full text.

The co-occurrence social network shows a smaller section of characters having strong connections. Visually, the network (Figure 5C) shows that the major protagonists and antagonists, like Krishna, are situated in the center and minor/supporting characters around the periphery, like that of the ground truth. However, like that of the word vector network from the full text, the central positioning of minor characters, such as Dhritarashtra, Drona, Duryodhana, and Sanjay, can be attributed to the fact that the nodes are arranged solely on how often a character "occurs" with another character, with highly co-occurring nodes being in the center. As such, frequently mentioned characters that are not necessarily as significant, such as those mentioned, are closer to the center.

**Table 2.** Performance of individual character-to-character relationships.

| Network | Vector Dimension | Category 1 | | | Category 10 | | | Category 15 | | |
|---|---|---|---|---|---|---|---|---|---|---|
| | | F | P | R | F | P | R | F | P | R |
| Full-text lw-KNN | 100 | 0.645 | 0.788 | 0.573 | 0.766 | 0.860 | 0.712 | 0.788 | 0.812 | 0.765 |
| | 200 | 0.676 | 0.817 | 0.600 | 0.784 | 0.878 | 0.725 | 0.909 | 0.938 | 0.882 |
| | 300 | 0.655 | 0.780 | 0.588 | 0.784 | 0.872 | 0.730 | 0.848 | 0.875 | 0.824 |
| | 400 | 0.655 | 0.770 | 0.594 | 0.783 | 0.866 | 0.735 | 0.812 | 0.812 | 0.812 |
| | 500 | 0.666 | 0.785 | 0.605 | 0.785 | 0.866 | 0.740 | 0.812 | 0.812 | 0.812 |
| | **600** | **0.667** | **0.785** | **0.607** | **0.785** | **0.866** | **0.740** | **0.812** | **0.812** | **0.812** |
| | 700 | 0.666 | 0.785 | 0.607 | 0.785 | 0.866 | 0.740 | 0.812 | 0.812 | 0.812 |
| | 800 | 0.663 | 0.782 | 0.603 | 0.789 | 0.873 | 0.743 | 0.812 | 0.812 | 0.812 |
| | 900 | 0.654 | 0.772 | 0.596 | 0.783 | 0.866 | 0.736 | 0.812 | 0.812 | 0.812 |
| | 1000 | 0.649 | 0.765 | 0.594 | 0.774 | 0.853 | 0.734 | 0.812 | 0.812 | 0.812 |
| Summary-text lw-KNN | 10 | 0.614 | 0.632 | 0.643 | 0.705 | 0.672 | 0.767 | 0.667 | 0.562 | 0.818 |
| | 20 | 0.616 | 0.611 | 0.674 | 0.681 | 0.634 | 0.77 | 0.714 | 0.625 | 0.833 |
| | 30 | 0.603 | 0.6 | 0.649 | 0.683 | 0.646 | 0.75 | 0.714 | 0.625 | 0.833 |
| | 40 | 0.593 | 0.591 | 0.641 | 0.671 | 0.632 | 0.748 | 0.759 | 0.688 | 0.846 |

**Table 2.** *Cont.*

| Network | Vector Dimension | Category 1 | | | Category 10 | | | Category 15 | | |
|---|---|---|---|---|---|---|---|---|---|---|
| | | F | P | R | F | P | R | F | P | R |
| | **50** | **0.589** | **0.591** | **0.628** | **0.672** | **0.632** | **0.744** | **0.759** | **0.688** | **0.846** |
| | 60 | 0.591 | 0.588 | 0.64 | 0.679 | 0.639 | 0.762 | 0.759 | 0.688 | 0.846 |
| Summary-text lw-KNN | 70 | 0.581 | 0.58 | 0.621 | 0.661 | 0.625 | 0.73 | 0.759 | 0.688 | 0.846 |
| | 80 | 0.581 | 0.58 | 0.621 | 0.661 | 0.625 | 0.73 | 0.759 | 0.688 | 0.846 |
| | 90 | 0.581 | 0.58 | 0.621 | 0.661 | 0.625 | 0.73 | 0.759 | 0.688 | 0.846 |
| | 100 | 0.581 | 0.58 | 0.621 | 0.661 | 0.625 | 0.73 | 0.759 | 0.688 | 0.846 |
| Co-occurrence | - | 0.615 | 0.777 | 0.567 | 0.709 | 0.739 | 0.717 | 0.741 | 0.625 | 0.909 |

Bold row values indicate the best performing setup.

### 3.2. Character-to-Character Relationships

Table 2 summarizes the F-score and related metrics of the social networks to capture the character-to-character relationships in the different character categories (category 1, category 10, category 15) under the various parameters. A three-way analysis of variance (ANOVA) with 10 (LSA word vector dimension) × 3 (character category) × 2 (corpus type) on the F-score metric showed that there were no significant interactions among the factors. However, the results showed that there was a significant difference between the character categories ($F(2,59) = 192.24$, $p < 0.001$) and there was a significant difference between the corpus types ($F(1,59) = 169.72$, $p < 0.001$). A Tukey post hoc test showed (all comparisons $p < 0.001$) that category 15 had a higher mean F-score ($\mu_{15} = 0.782$) than both category 10 ($\mu_{10} = 0.728$) and category 1 ($\mu_{1} = 0.626$). Additionally, a separate post hoc Tukey test showed that the full text had a higher mean F-score ($\mu_{Fulltext} = 0.755$) than the summary text mean ($\mu_{Summary} = 0.669$).

Although ANOVA could not be performed due to the degrees of freedom across the main three types of social networks and three character categories, the full-text-based networks had the highest mean F-score ($\mu_{Fulltext} = 0.755$), the co-occurrence network was second ($\mu_{Cooc} = 0.688$), and the summary text had the lowest ($\mu_{Summary} = 0.669$). Notably, the maximum F-score for the full text first occurred at a vector dimension of 600 in which F = 0.666 for category 1, F = 0.785 for category 10, and F = 0.812 for category 15. The maximum F-score for the summary text occurs at a vector dimension of 50, where F = 0.589 for category 1, F = 0.672 for category 10, and F = 0.759 for category 15. The co-occurrence analysis is only carried out once where F = 0.615 for category 1, F = 0.709 for category 10, and F = 0.741 for category 15.

Table 3 summarizes the F-score and related metrics of the social networks to capture the character-to-character relationships among the Pandavas or Kauravas groups. Due to the limitations of the degrees of freedom, an ANOVA could not be performed across all factors; thus, independent *t*-tests were performed to determine the statistical significance of the F-scores. For the Pandavas, a two-sample *t*-test ($t(18) = 22.80$, $p < 0.001$) showed that the mean F-score of the full-text lw-KNN generated networks ($\mu_{Fulltext} = 0.751$) was higher than the summary text lw-KNN networks, ($\mu_{Summary} = 0.645$). For the Kauravas, a two-sample *t*-test ($t(18) = 5.07$, $p < 0.001$) showed that the mean F-score of the full-text lw-KNN generated networks ($\mu_{Fulltext} = 0.585$) was higher than that of the summary text lw-KNN networks ($\mu_{Summary} = 0.551$).

**Table 3.** Performance of character-to-character relationships among the Pandavas and Kauravas.

| Network | Vector Dimension | Pandavas | | | Kauravas | | |
|---|---|---|---|---|---|---|---|
| | | F | P | R | F | P | R |
| Full-text lw-KNN | 100 | 0.747 | 0.814 | 0.708 | 0.562 | 0.766 | 0.463 |
| | 200 | 0.765 | 0.828 | 0.724 | 0.603 | 0.807 | 0.499 |
| | 300 | 0.748 | 0.809 | 0.711 | 0.579 | 0.757 | 0.488 |
| | 400 | 0.748 | 0.797 | 0.720 | 0.579 | 0.748 | 0.490 |
| | 500 | 0.749 | 0.790 | 0.730 | 0.598 | 0.781 | 0.502 |
| | **600** | **0.752** | **0.79** | **0.737** | **0.597** | **0.781** | **0.502** |
| | 700 | 0.752 | 0.790 | 0.737 | 0.596 | 0.781 | 0.500 |
| | 800 | 0.752 | 0.790 | 0.737 | 0.590 | 0.776 | 0.495 |
| | 900 | 0.752 | 0.790 | 0.737 | 0.574 | 0.758 | 0.480 |
| | 1000 | 0.747 | 0.782 | 0.737 | 0.569 | 0.751 | 0.478 |
| Summary-text lw-KNN | 10 | 0.665 | 0.613 | 0.758 | 0.573 | 0.647 | 0.548 |
| | 20 | 0.656 | 0.587 | 0.794 | 0.583 | 0.630 | 0.577 |
| | 30 | 0.664 | 0.602 | 0.781 | 0.553 | 0.599 | 0.541 |
| | 40 | 0.643 | 0.578 | 0.774 | 0.551 | 0.602 | 0.533 |
| | **50** | **0.633** | **0.578** | **0.737** | **0.554** | **0.602** | **0.538** |
| | 60 | 0.655 | 0.585 | 0.787 | 0.539 | 0.590 | 0.521 |
| | 70 | 0.633 | 0.578 | 0.735 | 0.538 | 0.582 | 0.528 |
| | 80 | 0.633 | 0.578 | 0.735 | 0.538 | 0.582 | 0.528 |
| | 90 | 0.633 | 0.578 | 0.735 | 0.538 | 0.582 | 0.528 |
| | 100 | 0.633 | 0.578 | 0.735 | 0.538 | 0.582 | 0.528 |

Bold row values indicate the best performing setup.

The maximum F-score for the full-text-based networks occurred at a vector dimension of 600, where F = 0.752 for the Pandavas and F = 0.597 for the Kauravas. The maximum F-score for the summary-text-based network occurred at a vector dimension of 50, where F = 0.633 for the Pandavas and F = 0.554 for the Kauravas.

### 3.3. Social Network Centralities

Table 4 summarizes the RMSE (lower is better) and correlation (higher is better) values of the four centralities of, (1) betweenness, (2) closeness, (3) degree, and (4) eigenvector that are extracted from the various constructed social networks and compared to the ground truth social network. The key results of each centrality are summarized below:

1.  Betweenness: The mean RMSE ($\mu_{Fulltext}$ = 0.162) of the full-text lw-KNN networks was lower than the RMSE ($\mu_{Summary}$ = 0.188) of the summary text lw-KNN generated social network ($t(18)$ = $-12.33$, $p < 0.001$). The full-text lw-KNN had the lowest error (RMSE = 0.159) at a vector dimension of 500 which was lower than the co-occurrence network (RMSE = 0.179) and summary text lw-KNN (RMSE = 0.183). The summary text lw-KNN had the first occurrence of its lowest at a vector dimension of 70.

2.  Closeness: The mean RMSE ($\mu_{Summary}$ = 0.033) of the summary text lw-KNN networks was lower than the RMSE ($\mu_{Fulltext}$ = 0.036) of the full-text lw-KNN social networks ($t(18)$ = 4.31, $p < 0.001$). The summary text lw-KNN had the lowest error (RMSE = 0.033) at a vector dimension of 70, full-text lw-KNN had the second lowest error (RMSE = 0.034) at a vector dimension of 500, and the co-occurrence had the worst error (RMSE = 0.037).

3.  Degree: The mean RMSE ($\mu_{Fulltext}$ = 0.063) of the full-text lw-KNN social networks ($\mu_{Fulltext}$ = 0.063) was lower than the RMSE ($\mu_{Summary}$ = 0.070) of the summary text networks ($t(18)$ = $-7.31$, $p < 0.001$). The full-text lw-KNN had the lowest (RMSE = 0.060) at a vector dimension of 500, followed by the co-occurrence network (RMSE = 0.065), and the summary text lw-KNN (RMSE = 0.068) at a vector dimension of 50.

4.  Eigenvector: The mean RMSE ($\mu_{Fulltext}$ = 0.069) of the full-text lw-KNN social networks is lower than the mean RMSE ($\mu_{Summary}$ = 0.076) of the summary text lw-KNN networks ($t(18)$ = $-6.08$, $p < 0.001$). The full-text lw-KNN had the lowest error

(RMSE = 0.066) at a vector dimension of 500, followed by the summary text lw-KNN (RMSE = 0.068) at a vector dimension of 50, and then the co-occurrence network (co-occurrence RMSE = 0.074).

Overall, across all the centrality measures, the full-text lw-KNN social network had the most highly correlated and statistically significant values as compared to the ground truth social network. The maximum correlation values for all centralities were highly significant with $p < 0.05$ and moderately positive [50], which is not the case for summary text lw-KNN or co-occurrence networks. Considering this, the full-text lw-KNN social network may be the best for capturing the relationships of the most critical characters to the story of the *Mahabharata* in a network structure.

**Table 4.** Comparison of constructed social network using centralities.

| Network | V. Dim. | Betweenness | | | Closeness | | | Degree | | | Eigenvector | | |
|---|---|---|---|---|---|---|---|---|---|---|---|---|---|
| | | RMSE | Corr. | $p$ | RMSE | Corr. | $p$ | RMSE | Corr. | $p$ | RMSE | Corr. | $p$ |
| Full-text lw-KNN | 100 | 0.166 | 0.399 | 0.08 | 0.036 | 0.405 | 0.08 | 0.065 | 0.445 | 0.05 | 0.075 | 0.351 | 0.13 |
| | 200 | 0.159 | 0.447 | 0.05 | 0.033 | 0.522 | 0.02 | 0.058 | 0.582 | 0.01 | 0.067 | 0.540 | 0.01 |
| | 300 | 0.162 | 0.442 | 0.05 | 0.036 | 0.444 | 0.05 | 0.062 | 0.523 | 0.02 | 0.068 | 0.500 | 0.02 |
| | 400 | 0.162 | 0.456 | 0.04 | 0.035 | 0.472 | 0.04 | 0.061 | 0.555 | 0.01 | 0.067 | 0.530 | 0.02 |
| | **500** | **0.159** | **0.470** | **0.04** | **0.034** | **0.509** | **0.02** | **0.060** | **0.571** | **0.01** | **0.066** | **0.544** | **0.01** |
| | 600 | 0.161 | 0.459 | 0.04 | 0.035 | 0.476 | 0.03 | 0.062 | 0.536 | 0.01 | 0.069 | 0.501 | 0.02 |
| | 700 | 0.163 | 0.445 | 0.05 | 0.035 | 0.476 | 0.03 | 0.062 | 0.533 | 0.02 | 0.069 | 0.501 | 0.02 |
| | 800 | 0.164 | 0.434 | 0.06 | 0.036 | 0.445 | 0.05 | 0.064 | 0.508 | 0.02 | 0.070 | 0.467 | 0.04 |
| | 900 | 0.164 | 0.433 | 0.06 | 0.036 | 0.441 | 0.05 | 0.064 | 0.499 | 0.03 | 0.071 | 0.454 | 0.04 |
| | 1000 | 0.165 | 0.434 | 0.06 | 0.037 | 0.421 | 0.06 | 0.066 | 0.470 | 0.04 | 0.073 | 0.419 | 0.07 |
| Sum.-text lw-KNN | 10 | 0.183 | 0.276 | 0.24 | 0.035 | 0.397 | 0.08 | 0.072 | 0.420 | 0.07 | 0.077 | 0.389 | 0.09 |
| | 20 | 0.202 | 0.197 | 0.41 | 0.035 | 0.406 | 0.08 | 0.075 | 0.419 | 0.07 | 0.081 | 0.347 | 0.13 |
| | 30 | 0.187 | 0.260 | 0.27 | 0.033 | 0.474 | 0.03 | 0.068 | 0.506 | 0.02 | 0.074 | 0.448 | 0.05 |
| | 40 | 0.191 | 0.259 | 0.27 | 0.033 | 0.466 | 0.04 | 0.071 | 0.484 | 0.03 | 0.078 | 0.402 | 0.08 |
| | **50** | 0.189 | 0.262 | 0.26 | 0.033 | 0.479 | 0.03 | **0.068** | **0.505** | **0.02** | **0.074** | **0.456** | **0.04** |
| | 60 | 0.191 | 0.254 | 0.28 | 0.033 | 0.465 | 0.04 | 0.070 | 0.487 | 0.03 | 0.074 | 0.453 | 0.04 |
| | **70** | **0.183** | **0.311** | **0.18** | **0.033** | **0.469** | **0.04** | 0.070 | 0.491 | 0.03 | 0.076 | 0.426 | 0.06 |
| | 80 | 0.183 | 0.311 | 0.18 | 0.033 | 0.469 | 0.04 | 0.070 | 0.491 | 0.03 | 0.076 | 0.426 | 0.06 |
| | 90 | 0.183 | 0.311 | 0.18 | 0.033 | 0.469 | 0.04 | 0.070 | 0.491 | 0.03 | 0.076 | 0.426 | 0.06 |
| | 100 | 0.183 | 0.311 | 0.18 | 0.033 | 0.469 | 0.04 | 0.070 | 0.491 | 0.03 | 0.076 | 0.426 | 0.06 |
| Co-occurrence | - | 0.179 | 0.355 | 0.12 | 0.037 | 0.451 | 0.05 | 0.065 | 0.516 | 0.01 | 0.072 | 0.466 | 0.04 |

Bold row values indicate the best performing setup.

### 3.4. Spectral Clustering Community Detection

Table 5 outlines the performance of the community detection provided by spectral clustering. A two-sample *t*-test ($t(18)$ = 4.35, $p < 0.001$) showed that the mean F-score of the full text fully connected networks ($\mu_{Fulltext}$ = 0.665) was higher than that of the summary text fully connected social networks ($\mu_{Summary}$ = 0.540). Notably, the F-score for the full text fully connected networks achieved the maximum at a vector dimension of 600 (F = 0.749) and the summary text fully connected networks reached a maximum at a vector dimension of 50 (F = 0.540). Figure 6 visualizes both of these two top performing spectral clustering setups.

**Table 5.** Performance of spectral clustering community detection of the Pandavas and Kauravas.

| Network | Vector Dimension | F | P | R |
|---|---|---|---|---|
| Full-text lw-KNN | 100 | 0.495 | 0.515 | 0.516 |
| | 200 | 0.520 | 0.581 | 0.625 |
| | 300 | 0.697 | 0.717 | 0.736 |
| | 400 | 0.697 | 0.717 | 0.736 |
| | 500 | 0.642 | 0.672 | 0.702 |

**Table 5.** *Cont.*

| Network | Vector Dimension | F | P | R |
|---|---|---|---|---|
| Full-text lw-KNN | **600** | **0.749** | **0.763** | **0.771** |
| | 700 | 0.749 | 0.763 | 0.771 |
| | 800 | 0.700 | 0.707 | 0.707 |
| | 900 | 0.700 | 0.707 | 0.707 |
| | 1000 | 0.700 | 0.707 | 0.707 |
| Sum.-text lw-KNN | 10 | 0.540 | 0.540 | 0.542 |
| | 20 | 0.500 | 0.505 | 0.505 |
| | 30 | 0.583 | 0.626 | 0.667 |
| | 40 | 0.540 | 0.571 | 0.583 |
| | **50** | **0.540** | **0.571** | **0.583** |
| | 60 | 0.540 | 0.571 | 0.583 |
| | 70 | 0.540 | 0.571 | 0.583 |
| | 80 | 0.540 | 0.571 | 0.583 |
| | 90 | 0.540 | 0.571 | 0.583 |
| | 100 | 0.540 | 0.571 | 0.583 |

Bold row values indicate the best performing setup.

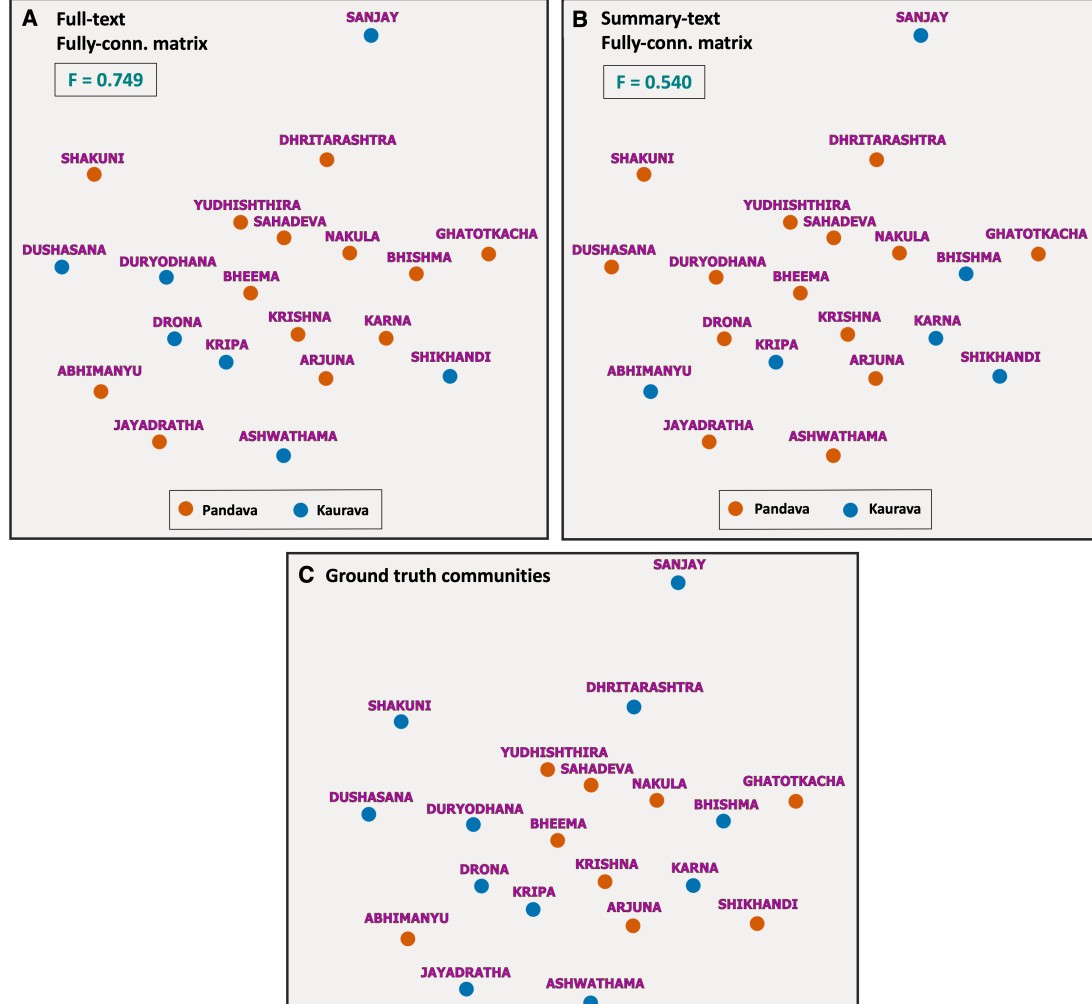

**Figure 6.** Spectral clustering detection of Pandavas and Kuravas. (**A**) Full text fully connected matrix at a word vector dimension of 600. (**B**) Summary text using the fully connected matrix at a word vector dimension of 50. (**C**) Ground truth memberships.

## 4. Discussion

The current study represents an important milestone in the digital humanities for research investigating non-Western literary work. To this end, we have applied modern NLP techniques and have developed methodological refinements to further enhance and quantitively verify insights that could be obtained from the famous South Asian literature, the (*Mahabharata*). First, we utilize contemporary word vector techniques based on matrix factorizations as a basis for constructing social networks. This is a novel procedure that contrasts with the standard co-occurrence analysis and other older procedures typically used [51,52]. This has allowed our social networks to incorporate a more nuanced embedding of the characters regarding their role in a literary work. Second, to further improve the efficacy of social networks to capture literary information, we introduced a new method of adaptively weighting the similarity among individual characters by computing the scaling parameter based on their relative importance in the text rather than using a global value for $\sigma$, which is typically the case. Third, we develop a locally weighted KKN graph which choose the $k$ neighbors for each character based on their frequency of occurrence in the *Mahabharata*. Again, the standard usage would be to have a single value of $k$ for all characters. Finally, we utilize spectral clustering for the task of community detection on literary works, which has not been previously investigated.

Co-occurrence analysis is one of the most popular methods used for social network extraction. For example, Das et al. [8] and Hutchinson et al. [53] use it as their primary procedure in constructing their respective social networks. However, different interaction detection methods such as conversations, affiliations and direct actions offer different strengths and have their own limitations [51]. Masías et al.'s work [52] investigates the characters of the Shakespearean play *Romeo and Juliet*. They implemented a modified version of conversation interaction detection (a subset of co-occurrence analysis) whereby they extract their characters using 'conversational turn-taking sequences'. This procedure focuses and tracks dialogues between characters with a 'who-talked-after-who' heuristic, which obtains a sequence of speaking turns of the characters. However, this technique ignores characters without dialogue and characters that are incidental to a plot and exclusively ascribes importance to talking roles. In contrast, word vectors are shown to embed a high amount of contextual information [18] which would include secondary, non-speaking and non-acting characters. Thus, our work mitigates the limitations of such interaction detection techniques by the utilization of the word vector methods.

Typically, interaction detection techniques are restricted to the previously mentioned methods of co-occurrence, conversations, etc. [51]. Our work and Hutchinson et al.'s [53,54] show that word vector methods, particularly LSA, are powerful alternatives for retrieving the same information. LSA, via SVD, was utilized in both works to create character word embeddings but was analyzed in different ways. Hutchinson et al. [53] applied LSA to a corpus of the complete *Harry Potter* series and visualized the word vectors of the characters on a two-dimensional plane using multidimensional scaling. In contrast, our methodology computes the similarity between the LSA word vector of the characters, which is then converted to a KNN graph/social network using the lw-KNN algorithm. Their findings show strong evidence for significant relational information being encoded throughout the text, which is not accessible by interaction detection alone, but can be detected with LSA [54]. They support this assertion with the high correlation between the relationship scores generated from the LSA and the corresponding actual relationships. However, their approach is limited in that they only generate relational information for single characters. Our work goes a step further in additionally generating inter-character relationships in a social network and using SNA to extract even subtler relational information such as significance and character hierarchies.

Many studies that implement social network analysis, including ours, utilize centrality measures as proxies for relational information such as significance and rank [7,38,41,55]. In using their first-order co-occurrence and higher-order co-occurrence, Hutchinson et al. [53,54] utilize a novel variant of degree centrality in their social network

computation. Similarly, Das et al. [8] calculates several centralities, including betweenness and degree, but as secondary measures to other metrics. In contrast, Masías et al. [52] clearly show that centralities can be used to make deep and important findings. They implement a weighted variant of the standard centralities to make important inferences about Juliet's role in the plot, reinterpreting it as much more significant than it is usually traditionally understood. Subsequently, our study utilized a wide range of centrality measures (betweenness, degree, closeness, and eigenvector), which allows for a greater breadth of character information. Traditionally, centrality measures have been used to indicate the social attributes of characters. However, the use of centralities as a quantitative way of comparing the fidelity of social networks (and the text they represent) is unexplored.

Original to our work, we used RMSE and correlation measures to compare the constructed network centralities to that of the ground truth network centralities. The RMSE and correlation scores indicate the degree to which the two mentioned social networks coincide, with a very low RMSE score and a significant correlation (*p*-value) suggesting that the constructed social network is very close to the ideal ground truth network. This has implications as a novel method for a text summarization procedure and as an alternative to the contemporary techniques [56–58]. This is particularly significant for the text summarization of fiction, which depends greatly on elements such as characters and social relations [59]. As such, this procedure can be utilized as a complementary metric for determining summary quality in addition to standard metrics such as ROUGE [60] and BLEU [61]. Specifically, this procedure can be used to make an evaluation of summaries of character-centric literary works by measuring the extent to which the character relations, and by extension, the total social dynamics, are preserved. For our work, this method suggests that the full-text lw-KNN method would be capable of preserving the character information and social interactions and, thus, can be adapted to or enhance text summarization of literary works.

The techniques implemented (e.g., LSA word vectors [21,62]) in this work are on par with that of contemporary NLP methods [63,64] and contrast with older studies such as Das et al. [8]. Yang [64] has shown that despite using contextual embeddings (such as BERT), which use state-of-the-art transformers and attention modules, their deep neural networks extracted character networks better in Jane Austen's *Sense and Sensibility* using static word embeddings such as Word2Vec and GloVe. These two word embeddings were previously shown to be equivalent to matrix factorization techniques such as SVD when the appropriate hyperparameters were used [20,21]. In their analysis of the *Mahabharata*, Das et al. [8] used a traditional part-of-speech (POS) tagging system for identifying characters in the text, which was then used with a standard co-occurrence analysis for generating the social network. We also replicated the same co-occurrence method, using the same co-occurrence window of one sentence, mainly as a comparison to the more advanced [65,66] word vectors/word embedding techniques presented in our study. Thus, we were able to show that compared to co-occurrence analysis, the word-vector-based procedures construct a much more idealistic and accurate social network of the characters.

None of the recent work in the Hindu literature has analyzed character networks as in-depth as our study. There have been recent studies analyzing the *Mahabharata*, but unfortunately they only look at word similarities derived from the Word2Vector word vectors and did not specifically analyze character networks [67,68]. Chandra and Ranjan [69] conducted a similar analysis on the *Upanishads* and *Bhagavad Gita* using BERT contextual embeddings for topic modelling, but again there was no analysis of the character network, as it is a difficult task to verify the quality of the character networks without annotating a ground truth network.

Overall, our study is more character-centric by utilizing centralities, spectral clustering, and direct character-to-character comparison, which goes beyond traditional studies [51,52,54]. In particular, community detection has not been prominent in computational literary analysis. We showed that community detection provided important social information concerning the sub-groups of characters that is not apparent using other techniques. Our analysis and

suite of methods allowed us to quantitatively uncover social information about the characters, the character communities, and character relationships. Nonetheless, some limitations were encountered in our research. Firstly, there is a level of subjectivity in the ground truths that typically cannot be eliminated [51] in a manual character annotation, such as that in the companion book. Secondly, while the bands utilized for the lw-KNN algorithm were experimentally validated, there is a margin of error associated with forcing a certain number of relations per character. However, these limitations will be a basis for our further investigations in developing techniques for automating the verification of social networks in literary works.

## 5. Conclusions

We demonstrate that word vectors can be used as an effective tool in the NLP analysis of literary works. In conjunction with a novel locally weighted KNN algorithm, it was shown that by utilizing this technique, we can construct accurate and representative social networks of characters in a literary artefact. With these methods, in our character analysis of the *Mahabharata*, the character relations and social dynamics of the characters in the epic are faithfully extracted. We also demonstrated that the validity of the social network can be evaluated by analyzing the character-to-character relations, the clusters generated by the community detection, and the centrality measures of the networks. Also, it was shown that the word vector methods applied to the full text produce significantly more accurate results when compared to those produced by word vectors on a summary text or co-occurrence analysis. The methods presented herein could also be used to assess whether summaries of literary works have captured the correct social relations during the summarization process. Finally, more non-Western literary sources should be studied to increase the cultural diversity represented in digital humanities. Specifically, a next step for South Asian literature NLP analysis could be the character analysis of the *Ramayana*.

**Author Contributions:** Conceptualization, E.G.; methodology, E.G.; software, V.M.; validation, V.M.; investigation, V.M.; resources, E.G.; data curation, V.M.; writing—original draft preparation, E.G. and V.M.; writing—review and editing, E.G.; visualization, E.G. and V.M.; supervision, E.G.; project administration, E.G. All authors have read and agreed to the published version of the manuscript.

**Funding:** This research received no external funding.

**Data Availability Statement:** The data presented in this study and code implementing the described methods will be available on the author's website.

**Conflicts of Interest:** The authors declare no conflict of interest.

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
