# Peer review of "A Quantitative Social Network Analysis of the Character Relationships in the Mahabharata"

_heritage, doi:10.3390/heritage6110366_

Round 1

Reviewer 1 Report

Comments and Suggestions for Authors

Everything is fine like abstract, introduction with clear contribution, with gooa analysis.

Reviewer 2 Report

Comments and Suggestions for Authors

This is an interesting paper which extends social analysis to non-English (or indeed European language texts, although the analysis was performed on an English translation.  The methods are quite traditional in this field, but the text studied is novel.  As such it is worthy of publication.

I did find the illustrations difficult to follow as there was insufficient contrast between the muddy background and yellow text.

Reviewer 3 Report

Comments and Suggestions for Authors

The authors conducted a social network analysis of the main characters in the Indian epic Mahabharata. The motivation of this paper is good, and the method proposed in this paper are clearly described.

However, I think the following issues should be further concerned.

(1) It will be good if you expand the explanation of  Old English  repository in section 2.2.

(2) Why SVD is used rather than other methods?

(3) The position of the figure is far from its citation. It is recommended to adjust the placement of the figures and cite them in the proper order.

(4) Please add the description of the algorithms in sections 2.3.1 and 2.4.3, and briefly explain their complexity.

(5) Please supplement the training method, loss function and other relevant descriptions for the lw-KNN algorithm.

(6) It will be good if you expand the explanation of "(F(2, 59) = 192.24, p < 0.001) " " (t(18) = 22.80, p < 0.001)" and other similar words  in section 3.2.

(7) Can lw-KNN algorithm be applied to other literary works? Has it been compared to other baseline models? please provide relevant data (if applicable) .

(8) The references listed are pretty old, and recent works should be added. 

Comments on the Quality of English Language

The quality of English language is acceptable.

Round 2

Reviewer 3 Report

Comments and Suggestions for Authors

I think the authors have improved the mauscript based on the reviewer's comments.

Comments on the Quality of English Language

The quality of English language is acceptable.